# Ultrastrong and tough paper structure from densified hybrids of multiscale cellulose fibers

Liqiong Liao [1], Bingbing Li [1], Zhiping Shi [1], Kai Li [1]✉, Yao Lu [1], Yuxin Liu[1] & Qi Zhou [2]✉

Replacing petrochemical plastics with sustainable materials requires strategies that combine high mechanical performance with low environmental impact. Here, we report an ultrastrong and tough cellulose paper created by integrating multiscale cellulose materials to overcome the weak fiber-to-fiber adhesion found in conventional paper. Bacterial cellulose microgels fill microscale voids between pulp fibers, while bacterial cellulose nanofibers reinforce nanoscale gaps, forming densely bonded interfaces through capillary-driven assembly during dehydration. This hybrid structure significantly enhances interfacial interactions and hydrogen bonding, producing cellulose paper with isotropic tensile strength of 811 MPa and robust wet-state performance. The process is highly energy-efficient due to rapid dewatering, consuming far less energy than nanocellulose-paper manufacturing. The resulting material also exhibits strong adhesion to hydrophilic substrates, expanding its application potential. By bridging the performance gap between natural cellulose fibers and traditional paper, this work provides a biodegradable, energy-efficient alternative to plastics for sustainable structural and packaging applications.

Driven by increased concerns about the long-term degradation of petrochemical plastics and the uncontrolled spread of microplastics, various paper-based packaging solutions with significantly improved mechanical performances have been developed as sustainable alternatives to non-degradable plastics[1–4]. It is important to note that individual wood pulp fibers, the fundamental elements for paper, possess remarkable yet often overlooked mechanical strength. Single wood fibers with low cellulose microfibril angle (MFA) have ultimate strength as high as 1000 MPa[5], while the ultimate tensile strength of a single wood pulp fiber with low MFA can reach 1700 MPa[6]. However, the tensile strength of paper fabricated from pulp fibers only reached 50 MPa, due to weak interfacial adhesion between the fibers[7–9]. To overcome this limitation, various strengthening agents have been introduced to enhance the interfacial interactions between the pulp fibers through chemical bonding, such as hydrogen bonds[10], electrostatic interactions[11], or covalent bonds[12,13]. Another approach was to increase the interfacial bonding area through the introduction of cellulose microfibrils (CMF) or cellulose nanofibers (CNF)[14,15]. Although these approaches could improve the strength of paper or board to some extent, a substantial gap remains when compared to the mechanical properties of individual pulp fibers.

Benefit from parallel packed load-bearing wood cells and naturally aligned cellulose microfibrils in wood cell walls, a two-step top-down process has been previously developed to fabricate strong wood cellulose films showing tensile strength of 449 MPa in the longitudinal direction. This process involves in-situ removal of lignin and hemicellulose, followed by a complete densification, facilitating the formation of a substantial interfacial area and ultimately stronger adhesion between naturally aligned wood cellulose fibers[16,17]. However, this strategy is not suitable for individualized and random-oriented

[1]Faculty of Chemical Engineering, Kunming University of Science and Technology, Kunming, China. [2]Division of Glycoscience, Department of Chemistry, School of Engineering Sciences in Chemistry, Biotechnology and Health, KTH Royal Institute of Technology, AlbaNova University Centre, Stockholm, Sweden. ✉e-mail: lkjnkmwh@hotmail.com; qi@kth.se

wood pulp fibers used in paper structures, as they are typically micron-scale in width (10 to 50 μm) and millimeter-scale in length (0.7 to 3 mm), and the voids within their network exhibit microscale dimensions. Nano-scale CMF and CNF materials could strengthen the joints between wood pulp fibers, but could not adequately fill the microscale voids between them. The hydrogen bonding, van der Waals force, and electrostatic interactions are short-range forces that only operate effectively when the spacing of material interfaces is below 300 Å[18]. The capillary forces, which are generated during the drying process in papermaking, are crucial for drawing the adjacent cellulosic fibers into close contact[19,20]. To achieve optimal mechanical properties, the microscale voids between the fibers must be filled to achieve a substantial densified interfacial surface and ensure effective interfacial interactions[21,22].

In this work, we utilized square-shaped bacterial cellulose microgels to fill the voids between wood pulp fibers. The 3D gel structure effectively interlocked with surrounding particles during the dehydration process, primarily driven by capillary forces facilitated by its porous architecture[23]. To enhance this effect, cellulose nanofibers were incorporated, generating stronger capillary forces due to the nanoscale distances between them, as predicted by the Young-Laplace equation[24]. These nanoscale cellulose fibers further filled the nanovoids in the paper structure. The combined use of micro- and nanoscale cellulose fibers effectively filled the gaps between wood pulp fibers, forming a tightly bonded interface driven by capillary forces. This configuration promoted the formation of additional cellulose-cellulose and cellulose-bound water hydrogen bonds, ultimately resulting in a paper structure of superior mechanical properties at both dry and wet states. Furthermore, the adhesive performance of the hybrid of multiscale cellulose fibers for substrates of hydrophilic nature was demonstrated.

## Results

### Hybrid of cellulose microfibers and nanofibers on paper strength

Hybridization of fibers with different sizes, such as carbon fibers, nanofibers, and nanotubes, has been widely applied in fiber-reinforced cementitious and polymer composites to achieve higher strength, modulus, and fracture toughness[25–28]. Fibers of smaller dimensions effectively bridge micro-cracks, whereas larger fibers are designed to inhibit the propagation of macro-cracks within the composite material. Herein, the preparation, structure, and properties of the hybrid paper structures of cellulose fibers, microfibers, and nanofibers are explored. As illustrated in Fig. 1a, millimeter-long spruce wood pulp fibers (Fig. 1b), microscale cellulose microgel (average size of 72.9 μm) obtained from bacterial cellulose (Fig. 1c and Supplementary Fig. 1), and TEMPO-oxidized bacterial cellulose nanofibers (B-CNF, Fig. 1d) were mixed in various ratios to prepare the hybrid cellulose papers (HCPs) using a standard papermaking procedure. The impact of the compositions of pulp fibers, microgel, and B-CNF on the mechanical properties of the resulting HCPs was studied. Notably, the HCP composed of pulp fibers (65° SR), microgel, and B-CNF in a 1:1:1 ratio exhibited high mechanical properties, particularly a synergistic enhancement in modulus, tensile strength, and strain-to-failure compared to the neat fibers (Fig. 1e). The HCP showed an exceptionally high ultimate tensile strength of 811 MPa and a high modulus of 36.1 GPa, surpassing all random-in-the-plane oriented (isotropic) cellulose papers or films reported to date (Fig. 1f). These values, particularly a specific tensile strength of $7.37 \times 10^{-1}$ MPa m³/kg, a specific Young's modulus of $3.28 \times 10^{-2}$ GPa m³/kg, a toughness of 56.5 MJ/m³ are comparable and even partially higher than those of strong cellulose films with cellulose microfibrils or wood fibers aligned-in-the-plane[17], super aligned bacterial cellulose (BC) films[29], and of anisotropic nanofibers-structured regenerated cellulose films[30] (Supplementary Table 1). Interestingly, the HCP showed a strain hardening behavior

during tensile testing (insert picture in Fig. 1e and Supplementary Movie 1), similar to the plastic deformation of elastomeric polymers. Notably, upon unloading after the onset of strain hardening but prior to fracture, the sample retained its deformed (homogeneous necking) shape, confirming the presence of plastic deformation (Supplementary Fig. 2). This phenomenon was previously reported in cellulose nanopaper prepared from microfibrillated cellulose (MFC), which showed a yield stress of 91 MPa and a slope of 1.82 GPa in the plastic region. In comparison, the HCP sample demonstrated remarkably enhanced mechanical performance, with a yield stress of 305 MPa, a plastic deformation slope of 4.76 GPa, and a high strain-to-failure of 11.4%. These values suggest significantly increased frictional resistance to fiber slippage within the network and superior toughness resulting from the hybridization of multiscale cellulose fibers. To elucidate the failure mechanisms underlying this behavior, the fracture surface of the HCP was examined using scanning electron microscopy (SEM, Supplementary Fig. 3). Two dominant fracture modes were identified: fiber pull-out and fiber rupture. The presence of ruptured fibers confirms that load transfer through the network reached a threshold sufficient to cause fiber breakage, enabling greater energy absorption compared to pull-out alone. This mixed-mode fracture behavior is a key contributor to the enhanced mechanical properties observed in the HCP.

We thoroughly investigated the mechanical properties of papers prepared from ternary hybrids of pulp fibers, microgel, and B-CNF at different ratios (Fig. 1g). The results revealed that small adjustments in composition (e.g., 1.2:1:1 or 0.8:1:1) could not lead to further improvements compared to the HCP at 1:1:1. The hybrid papers with higher content of pulp fibers (2:1:1) or microgel (1:2:1) showed higher strain-to-failure but significantly lower modulus and tensile strength. The HCP with a composition of 1:1:1 remarkably outperformed paper with other compositions in terms of Young's modulus, tensile strength, and toughness (Supplementary Table 2). In addition, the beating degree of the pulp fibers has a strong impact on the mechanical properties of HCPs, which initially improved and then declined with increasing beating degree of the pulp fibers (Fig. 1h), as determined by the Schopper-Riegler (°SR) scale. HCPs fabricated with pulp fibers at beating degrees of 45 °SR, 65 °SR, and 75 °SR exhibited tensile strengths of 586 MPa, 811 MPa, and 538 MPa, respectively. This trend mirrors the effect of beating degree on the mechanical properties of traditional paper[31], highlighting the dominant role of pulp fiber condition in determining the mechanical performance of HCPs. The initial increase in beating degree promoted fiber fibrillation, enhancing interfacial interactions between the pulp fibers and improving mechanical strength. However, excessive beating caused excessive fiber shortening and fines generation, ultimately reducing mechanical performance.

The respective roles of microgel and B-CNF in enhancing paper strength were evaluated using composite papers prepared from binary combinations of B-CNF and pulp fiber, microgel and pulp fiber, and microgel and B-CNF (Supplementary Fig. 4). When combined with pulp fibers, B-CNF led to a modest increase in tensile strength and stiffness, attributable to its nanofibrillar structure and reinforcing effect. In contrast, the incorporation of microgel into B-CNF networks resulted in increased strain-to-failure, albeit at the expense of strength and stiffness. Notably, when microgel was combined with pulp fibers, it enhanced interfacial adhesion and contributed to a synergistic improvement in both tensile strength and toughness.

The influence of microgel average particle size on the mechanical properties of the ternary hybrids was further investigated. A reduction in particle size from 72.9 to 52.1 and 13.5 μm did not alter the overall shape of the stress-strain curve, but it led to a decrease in both strain-to-failure and ultimate tensile strength (Supplementary Fig. 5). This trend is analogous to the mechanical performance deterioration observed with decreasing average molar mass of

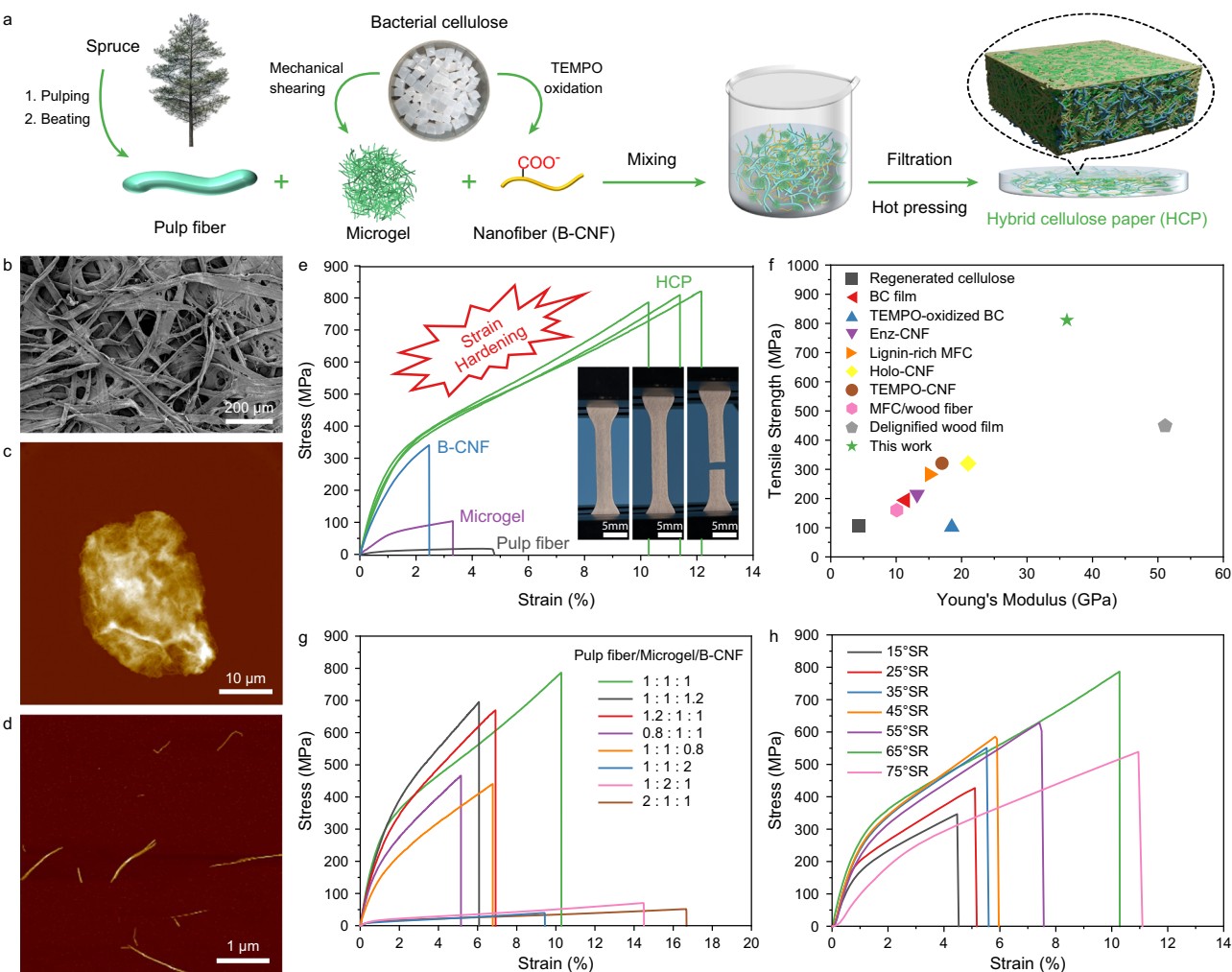

**Fig. 1 | Role of multiscale cellulose fiber hybrids in enhancing paper strength.**
**a** Schematic of the general preparation route for hybrid cellulose paper (HCP) using three types of cellulose fibers with different size scales. (**b**) SEM image of the pulp fibers (65° Schopper-Riegler, SR). **c** AFM image of the cellulose microgel. **d** AFM image of the cellulose nanofibers (B-CNF). **e** Tensile stress–strain curves of the HCP prepared from the pulp fiber, microgel, and B-CNF with a composition of 1:1:1 by weight as compared to the papers prepared from the individual components; the inset photograph shows strain hardening of the HCP under tensile loading. **f** Comparison of the mechanical properties of the HCP developed in this work with various random oriented (isotropic) cellulose-based papers or films, including regenerated cellulose[30], bacterial cellulose (BC) film[29,45], TEMPO-oxidized BC[46], enzyme-treated CNF (Enz-CNF)[32], lignin-rich microfibrillated cellulose (MFC)[47], holocellulose CNF (Holo-CNF)[36], TEMPO-oxidized CNF (TEMPO-CNF)[37], MFC/wood fiber composite[48], delignified wood film[17], in a tensile strength versus Young's modulus plot. **g** Tensile stress–strain curves of HCPs fabricated from different weight ratios of pulp fibers, microgel, and B-CNF. **h** Tensile stress–strain curves of HCPs prepared using pulp fibers with varying beating degrees, maintaining a constant 1:1:1 weight ratio of the three fiber components.

cellulose[32]. Furthermore, mechanically homogenized MFC, derived from bleached softwood kraft pulp and characterized by widths of 10–20 nm and lengths of 5–10 μm, was used in place of B-CNF, which has a carboxylate content (1.18 mmol/g) and dimensions of 5–12.5 nm in width and 1–2.3 μm in length. Although the microgel/MFC combination showed a moderate synergistic effect compared to individual components (Supplementary Fig. 6), the substitution resulted in HCP samples with lower yield stress (100 MPa) and ultimate tensile strength (210 MPa). These findings emphasize the critical role and synergistic interaction between thin nanoscale cellulose fibrils (B-CNF) and microscale cellulose microgel particles in governing the mechanical performance of the HCP.

## Structure of hybrid cellulose paper

The structure of the HCP with remarkable mechanical properties was thoroughly investigated. The drying process of the hybrid multiscale fibers is illustrated in Fig. 2a and characterized by rheological characterizations and SEM analysis. The pulp fibers, microgel, and B-CNFs initially assembled into a gel-like dispersion (Supplementary Figs. 7 and 8). After vacuum filtration and drying at 105 °C, the wet hybrid fiber web formed a densified paper structure with a bulk density of 1.10 g/cm³ and a porosity of 27% (Supplementary Table 3). The cross-sectional SEM images of the papers fabricated from the neat pulp fibers (Fig. 2b), a 1:1 mixture of the pulp fibers and B-CNF (Fig. 2c), and a 1:1 mixture of the pulp fibers and microgel (Fig. 2d) showed a more porous, typical multilayered paper structure with distinct micrometer-scale gaps between layers and voids of between pulp fibers. In contrast, the cross-sectional SEM images of the HCP (Fig. 2e) revealed a compact, continuous, multilayered structure formed by the hybridization of the pulp fibers, microgel, and B-CNF, with the gaps and voids filled by the microgel and B-CNF. As shown in Supplementary Table 3, the HCP paper exhibited porosity comparable to that of neat B-CNF and microgel papers yet demonstrated markedly superior mechanical performance (Fig. 1e). This enhancement suggests strong interfacial adhesion and improved interfacial toughness arising from the synergistic hybridization of pulp fibers with B-CNF and microgel.

The crystalline structures of the HCP and papers prepared from neat pulp fiber, microgel, and B-CNF were characterized using Wide-

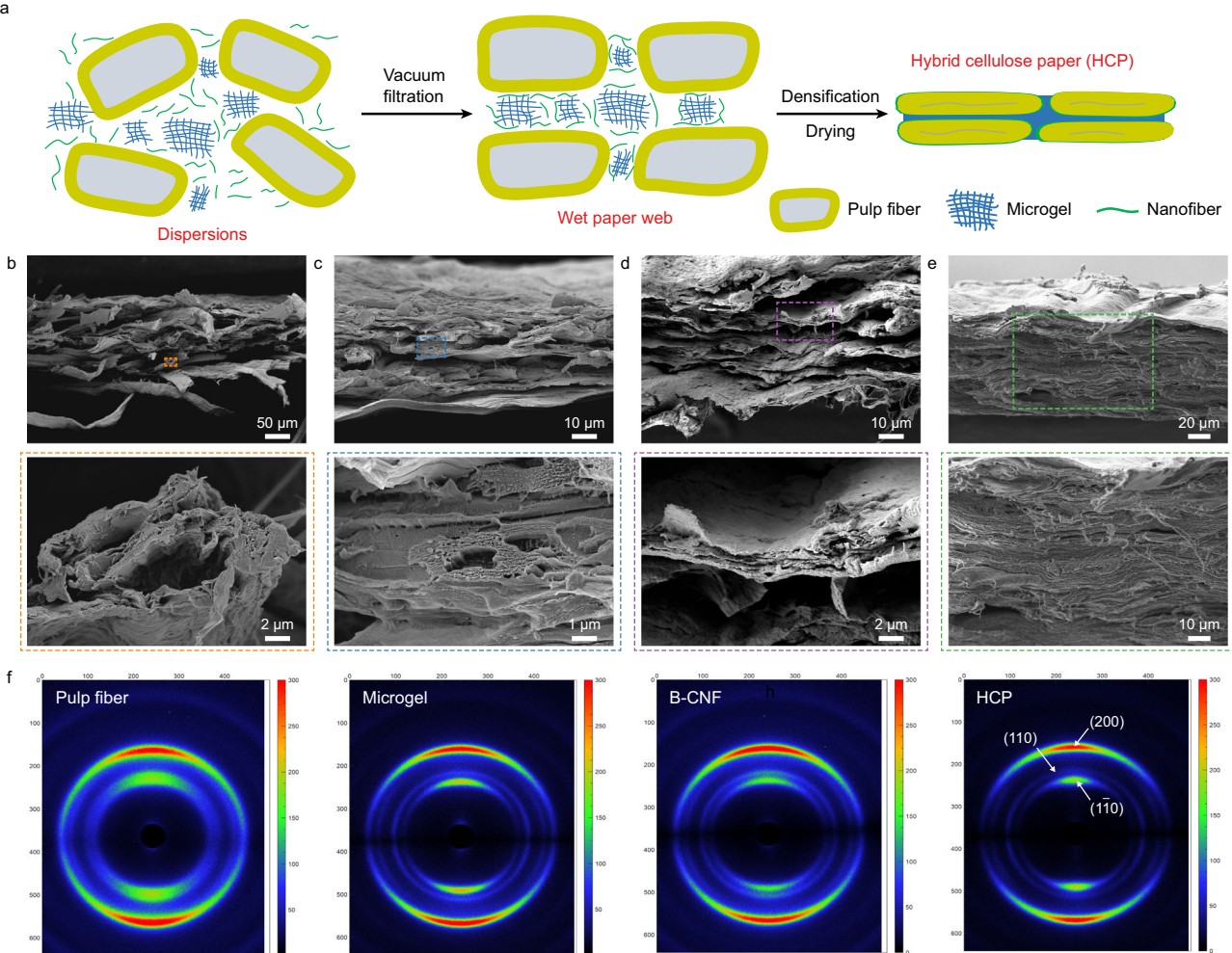

**Fig. 2 | Structural characteristics of hybrid cellulose paper (HCP). a** Schematic illustration showing the structural evolution from a fiber dispersion to a wet paper web formed by vacuum filtration, followed by densification into HCP after drying. Cross-sectional surface SEM images of the papers prepared from **b** neat pulp fiber (65° SR), **c** a hybrid of pulp fibers (65° SR) and B-CNF (1:1 ratio), **d** a hybrid of pulp fibers (65° SR) and microgel (1:1 ratio), and **e** a ternary hybrid of pulp fiber (65° SR), microgel, and B-CNF (1:1:1 ratio). **f** Wide-angle X-ray scattering (WAXS) diffractograms with the incident beam parallel to the paper surface of the HCP and control samples.

angle X-ray diffraction (WAXS). When the X-ray beam was perpendicular to the paper surface as the X-ray diffractograms shown in Supplementary Fig. 9, all papers showed typical ring patterns of cellulose I crystals in the paper surface plane, indicating random-in-the-plane distribution of cellulose fibrils in all papers. The structure in the cross-sectional plane was revealed with the X-ray beam parallel to the paper surface (Fig. 2f). In contrast, all papers showed a pair of bright arcs corresponding to the $(1\bar{1}0)$, (110), and (200) diffractions of cellulose I crystal, indicating cellulose fibrils were aligned parallel to the paper surface as a result of vacuum filtration in the papermaking process. The cellulose crystal orientation was quantified using the (200) reflection to evaluate the degree of parallel packing of the multiscale cellulose fibers. The HCP sample had an orientation factor ($f_c$) value of 0.83 higher than those of papers from neat pulp fibers ($f_c$ of 0.54), microgel ($f_c$ of 0.57), and B-CNF($f_c$ of 0.63), suggesting improved dense parallel packing of cellulosic fibers as also observed by SEM analysis (Fig. 2e). The crystallinity indices of the pulp fiber, microgel, B-CNF, and the HCP were measured to be 55.9%, 65.5%, 67.6%, and 63.5%, respectively (Supplementary Fig. 10). These results indicate that the hybridization of multiscale cellulose fibers did not significantly alter the overall crystallinity of the composite.

Capillary force is a long-range force that can draw multiscale cellulose fibers together over larger distances, while hydrogen

bonding becomes effective at distances below 300 Å. It can be inferred that when the distance between the surfaces of cellulose fibers is greater than 300 Å, capillary force plays a dominant role, pulling the fibers closer. Once the distance is less than 300 Å, hydrogen bonding, combined with capillary force, further pulls the fibers together, forming more hydrogen bonds and resulting in a dense structure. As referenced, there are two types of hydrogen bonds in cellulose materials: those between cellulose molecules and those between bound water and cellulose. To determine which type of hydrogen bonding primarily influences mechanical properties, the HCP was completely dried under vacuum at 110 °C for approximately 2 to 3 h to remove bound water. The NIR spectra (Fig. 3a) revealed a significant difference between HCP subjected to complete drying and natural drying, marked by the absence of the peak at 5178 cm$^{-1}$, which corresponds to bound water[33]. To determine the hydration state of the bound water in the HCP sample, low field nuclear magnetic resonance (LF-NMR) measurements were performed at room temperature (Fig. 3b). The natural dried HCP sample had a moisture content of 5.8% under 50% relative humidity (RH) and showed only a strong peak at $T_2 < 1$ ms, corresponding to tightly bound water that was mostly located in the first hydration layer, where water binds to the hydroxy groups of cellulose. When the HCP samples were conditioned under 70% and 90% RH, their moisture contents increased to 10.8% and 18.5%, respectively.

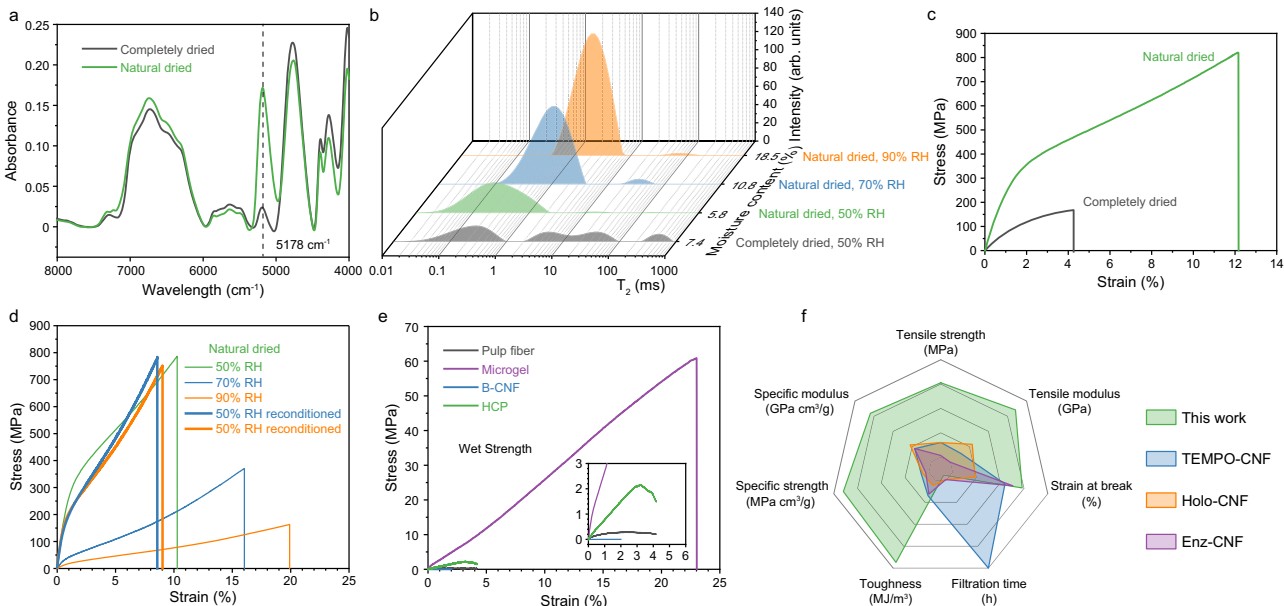

**Fig. 3 | Influence of bound water on mechanical performance of the hybrid fibers. a** Near-infrared (NIR) spectra of the HCP paper after complete and natural drying. **b** $T_2$ inversion spectra for the HCP prepared by natural and complete drying and conditioned at different relative humidity (RH) levels. **c** Representative stress–strain curves of the HCP under both drying conditions. **d** Typical stress–strain curves of the natural dried HCP at different RH levels. **e** Stress–strain behavior of the wet HCP and papers prepared from individual fiber components after incubation in water for 24 h. **f** Radar plots comparing mechanical properties and filtration times of the HCP (this work) with those of TEMPO-CNF, Holo-CNF, and Enz-CNF.

Correspondingly, the $T_2$ relaxation times of the bound water shifted to values above 1 ms and exhibited broader distributions, indicating a greater contribution from more loosely bound water under high humidity conditions. Additionally, small relaxation peaks appeared at 70 ms and 500 ms, which suggests the presence of a minor fraction of free or bulk-like water at high RH levels. After complete drying, the HCP sample had a moisture content of 7.4% under 50% RH and showed a significantly decreased peak for tightly bound water ($T_2 < 1$ ms) and 3 additional peaks with $T_2$ time of 5 ms, 40 ms, and 500 ms, respectively. These peaks correspond to loosely bound water that is located not far away from hydroxy groups, as well as free water confined in the pores due to capillary condensation and bulk-like water resulting from surface condensation, respectively. These results indicate the irreversible loss of tightly bound water in HCP after completely drying. The mechanical properties of HCP declined drastically after complete drying, with tensile strength decreasing from 811 MPa to 167 MPa (Fig. 3c). Furthermore, when the RH was increased to 70% and 90%, the ultimate tensile strength of the natural dried HCP decreased significantly while the strain-to-failure increased (Fig. 3d), consistent with the higher moisture content and the greater proportion of loosely bound water (Fig. 3b). Notably, the tensile strength of the natural dried HCP was largely restored after reconditioning the samples back to 50% RH (Fig. 3d), indicating that the humidity-induced changes were reversible. These observations indicate that tightly bound water, which is hydrogen-bonded to the cellulose chains[34,35], plays a key structural role in HCP. This tightly bound water contributes to the dense packing and load transfer within the network of multiscale cellulose fibers and is therefore a primary factor underlying the material's ultrahigh strength and toughness.

On the other hand, we introduced free water into the HCP and papers prepared from neat individual components, as well as binary and ternary hybrids with different compositions. The coefficient of water absorption for the neat pulp fibers, microgel, and their hybrids is at least 5-fold lower than that of neat B-CNF and hybrids containing B-CNF (Supplementary Fig. 11). This is attributed to the carboxyl groups generated by TEMPO-mediated oxidation in B-CNF, which

ionize in water, leading to electrostatic interactions that disassemble the dense stacking of cellulose materials. Correspondingly, the mechanical properties of the HCP and the paper prepared from neat B-CNFs and pulp fibers significantly decreased after immersed in water for 24 h (Fig. 3e). The wet tensile strength of the HCP was 1.3 MPa, while the cellulose microgel exhibited higher wet strength of 62 MPa due to lower water absorption, suggesting the interlocking structure of the cellulose microgel is robust enough to resist disassembly and water infiltration.

The high energy consumption typically associated with vacuum filtration during the fabrication of nano-sized or micro-sized cellulose materials has often limited their practical applications. The energy consumptions for the fabrication of HCP and papers from the three neat component cellulose fibers were evaluated. The vacuum filtration time for the HCP was 1.417 h, significantly lower than the 7.79 h required for B-CNF, highlighting its great potential for practical applications (Supplementary Fig. 12). Moreover, the HCP exhibited superior mechanical properties to cellulose films prepared from TEMPO-CNF[36,37], Holo-CNF[36], and Enz-CNF[32,38], while also achieving a shorter filtration time and better energy savings, as shown in Fig. 3f and Supplementary Table 4.

### Adhesive performance of the hybrid cellulose fibers

To further understand the underlying mechanism responsible for the formation of dense structure from the pulp fiber (65° SR), microgel, and B-CNF, the adhesive properties of the hybrid multiscale cellulose fibers were investigated. We hypothesize that the driving forces for the formation of the wet paper web and the natural drying process are primarily attributed to capillary forces and hydrogen bonding between bound water and cellulose fibers. Both driving forces are related to interfacial interactions, which effectively pull the multiscale cellulose fibers together. We demonstrated the difference of adhesion mechanism between B-CNF and microgel using glass slides as the substrate ss shown in Fig. 4a. The B-CNF formed scattered cellulose nanofibrils aggregates between the glass substrates due to fractures occurred in the gelled B-CNF suspension upon drying similar to

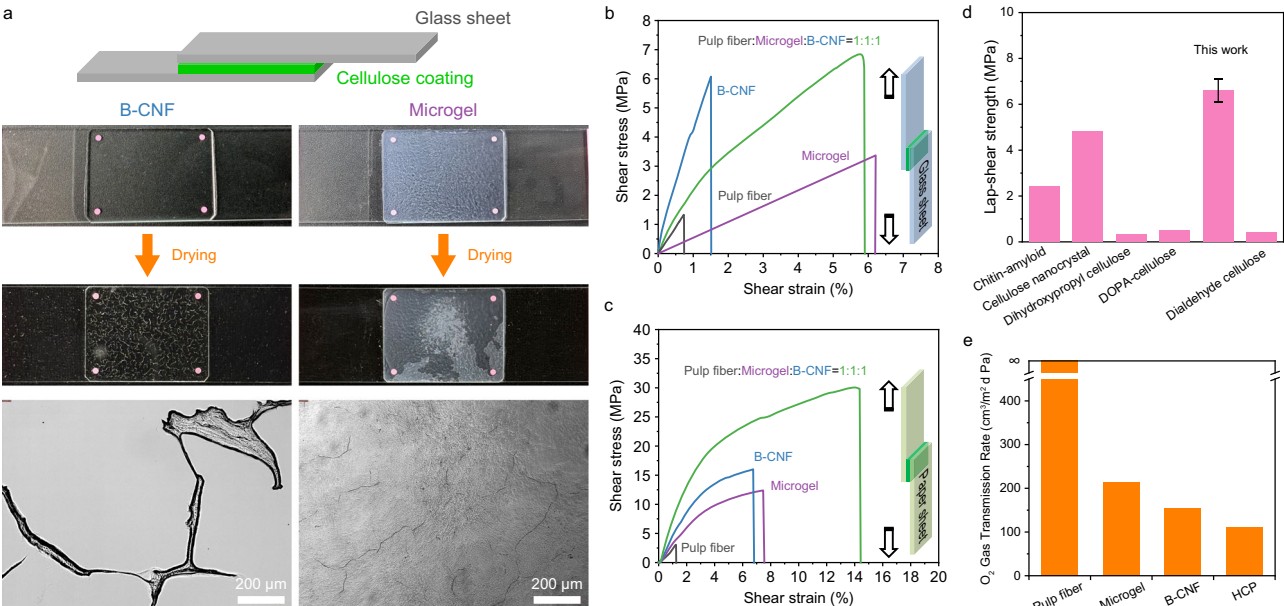

**Fig. 4 | Adhesive performance of the hybrid cellulose fibers. a** Comparison of adhesive mechanism between B-CNF and microgel for hydrophilic substrates (glass slides). 10 mg of B-CNC or microgel suspended in 1 mL of water were left to assemble between two glass plates and heated at 105 °C for 12 h. The bonded area between two glass slides was observed in bright field (scale bar: 200 μm). **b** Lap-shear stress–strain curves of glass substrates adhered using different cellulose fiber dispersions and their hybrids. **c** Lap-shear stress–strain curves of paper substrates bonded with the same set of cellulose-based materials. **d** Comparison of the lap-shear strength of glass substrates adhered using the HCP with other representative polysaccharide-based adhesives. Data are presented as mean values ± standard deviation (SD), based on $n = 5$ independent replicates. **e** Oxygen transmission rate of the HCP compared to the papers made from neat pulp fiber (65° SR), neat microgel, and neat B-CNF.

cellulose nanocrystals[39], while cellulose microgel formed a homogeneous nanofibril network without fractures owing to the high internal cohesion in the cellulose microgel upon drying. We further investigated the interfacial adhesion of dispersions of the pulp fiber (65° SR), microgel, and B-CNF, along with their binary mixtures and ternary mixture, on glass (Supplementary Fig. 13) and paper (Supplementary Fig. 14) substrates. Remarkably, the cellulose hybrid containing pulp fibers (65° SR), microgel, and B-CNF in a 1:1:1 ratio exhibited strong adhesion with a lap-shear strength of $6.6 \pm 0.5$ MPa and $27.0 \pm 1.1$ MPa for the glass and paper substrates, respectively, significantly higher than those of the individual cellulose fiber components (Fig. 4b, c). The lap-shear strength of the hybrid cellulose fibers adhered to the glass sheet also surpasses that of other polysaccharide-based materials (Fig. 4d), including chitin-amyloid[40], cellulose nanocrystal[39], dihydroxypropyl cellulose[41], DOPA-cellulose[42], and dialdehyde cellulose[43]. Furthermore, the adhesive behavior of the hybrid cellulose fibers for paper sheets closely resembled its behavior in tensile tests (Fig. 1e), exhibiting synergistic higher modulus, strength, and strain compared to its individual components (Fig. 4c). These findings confirm that the enhanced toughness of the HCP primarily arises from the interfacial interactions among the three cellulose fibers of different sizes. The resulting densely packed microstructure not only governs the mechanical response but also imparts high gas-barrier performance with an oxygen transmission rate of 110.2 cm³/m² day Pa (Fig. 4e).

## Discussion

In summary, we fabricated an ultrastrong and tough cellulose paper structure by harnessing the inherent mechanical properties of traditional pulp fiber. Micro-sized cellulose microgel and nano-sized B-CNF were utilized as interfacial adhesives for pulp fiber, resulting in the strongest unoriented cellulose paper to date, with a tensile strength of 811 MPa in all directions. The factors underlying this exceptional

mechanical performance were systematically investigated. The hybridization of pulp fiber (65° SR), B-CNF, and microgel formed a dense, multi-layered stacking structure. To evaluate the interfacial interactions and adhesion between the pulp fibers, microgel, and B-CNF, the ternary hybrid was tested as an adhesive for paper and glass substrates, exhibiting stronger adhesion than any of its individual components. Furthermore, the nature of these interactions was elucidated, with hydrogen bonding between bound water and cellulose identified as a key contributor to the high mechanical properties of the HCP. Moreover, the energy consumption during the vacuum filtration of HCP was found to be significantly lower compared to B-CNF alone. This work paves the way for advancing the application of traditional pulp fibers in the development of high-performance, energy-efficient cellulose-based structural materials.

## Methods

### Materials
Spruce wood pulp fibers were obtained from Jiulong Pulp & Paper Co., Ltd. (China). Sodium chlorite ($NaClO_2$, 80%), sodium hypochlorite (NaClO), sodium phosphate, and 2,2,6,6-tetramethyl-1-piperidinyloxy (TEMPO) were purchased from Sigma-Aldrich (Germany). Bacterial cellulose (BC) cubes ($1 \times 1 \times 1$ cm³) were sourced from Yeguo Co. Ltd. (China). The BC cubes were homogenized to fabricate the microgel using a high-speed blender (HR3865, Philips, China) at 42,000 rpm for 20 min. The mean particle size of the obtained microgel was 72.9 μm. To reduce the particle size, the microgel dispersion was further homogenized for 5 and 20 min using a colloid grinder (MX-JTM-G25, Mixpower, China). The microgel suspension had a solid concentration in the range of 0.40–0.65 wt.%.

### Preparation of B-CNF
BC cubes (1 g) were suspended in 100 mL water containing TEMPO (0.016 g, 0.1 mmol) and sodium bromide (0.1 g, 1 mmol). The TEMPO-

mediated oxidation was initiated by adding the desired amount of the NaClO solution (equivalent to 10 mmol NaClO per gram of cellulose) and was carried out at room temperature with continuous stirring at 500 rpm. The pH of the reaction was maintained at 10 by adding 0.5 M NaOH solution. After NaClO was consumed, the TEMPO-oxidized BC fibers were washed thoroughly and disintegrated using a high-speed blender (HR3865, Philips, China) at 42,000 rpm for 10 min to obtain B-CNF. The resulting B-CNF water suspension had a solid concentration in the range of 0.2–0.3 wt.%.

### Preparation of hybrid cellulose paper
Multiscale cellulose fibers, including spruce wood pulp fibers, microgel, and B-CNF, were combined in various ratios and mixed at 500 rpm for 10 min. The pulp fiber suspension in water had a solid concentration of 10 wt.%. After mixing pulp fiber, microgel, and B-CNF, the final hybrid suspension was adjusted to a solid concentration of 0.17 wt.%. The hybrid cellulose papers were then prepared using a standard procedure, which involved vacuum filtration followed by drying in a sheet former (Model IMT-GZ01) at 105 °C for 60 min.

### Lap-shear adhesion
Various dispersions of multiscale cellulose fibers hybrid were applied to glass or paper substrates (dimensions: length × width = 12 cm × 2.5 cm) with a coating area of 2.5 cm × 2.5 cm and a constant solid content of 10 mg. The dispersions were applied at a solid concentration of approximately 1.5–2.0 wt.%, achieved by concentrating the hybrid suspension by rotary evaporation. The substrates were then transferred to a sheet former and dried at 105 °C for 60 min. The paper substrate was prepared from the spruce wood pulp fibers (65°SR) following the same procedure as the hybrid cellulose paper, with a thickness of 0.1 mm. The thickness of the glass slides was 4 mm.

### Characterizations
The particle size distribution of the cellulose microgel dispersions was measured by a laser diffraction particle size analyzer (S3500, Microtrac, USA). The tensile mechanical properties of the samples were measured using an Instron 5944 tensile tester equipped with a 500 N load cell. The lap-shear adhesion tests were performed using a universal testing machine (UTM-2502) at a tensile speed of 2 mm/min. The sample morphology was characterized using scanning electron microscopy (SEM) with a GeminiSEM 460 instrument operating at an accelerating voltage of 5.0 kV. Wide-angle X-ray scattering (WAXS) data were collected using a Xeuss 2.0 instrument equipped with a curved detector (Xenocs Corp.) and CuKα radiation ($\lambda$ = 0.1541 nm) at an operating voltage of 40 kV and a current of 30 mA. Two-dimensional diffraction patterns were recorded by mounting the samples either perpendicular or parallel to the incident beam. Near-infrared (NIR) spectra of the HCP films were collected using an FT-NIR spectrometer (MPA II, Bruker, Germany) equipped with an InGaAs detector and a diffuse reflectance integrating sphere (60 mm diameter) at 25 °C and 50% RH. Spectra were acquired in the wavelength range of 8000–4000 cm$^{-1}$ with 1024 scans at a spectral resolution of 8 cm$^{-1}$. All spectra were normalized using the min–max normalization method, implemented in the OPUS software. Gas barrier properties of the samples were evaluated according to the GB/T 1038.1-2022 standard method using a Basic201 differential pressure gas permeameter. Circular samples (diameter: 9 cm) were cut using a mold and placed in the test cell, and then degassed under vacuum prior to testing. Nuclear magnetic resonance (NMR) relaxation measurements were conducted using a low-field pulsed NMR imager and analyzer (NMI20, Niumag Corporation, Shanghai, China) operating at a resonance frequency of 18 MHz. One gram of pretreated cellulose membrane powder was placed in a glass tube (15 mm inner diameter) and inserted into the

NMR probe, which was maintained at a constant temperature of 32 °C. The transverse spin-spin relaxation time ($T_2$) was measured using the Carr-Purcell-Meiboom-Gill (CPMG) pulse sequence. The 90° and 180° pulse durations were 14 µs and 27 µs, respectively, with an inter-pulse delay of 200 µs. A total of 18,000 echoes (data points) were collected per measurement. To improve the signal-to-noise ratio, eight scans were averaged using a repetition time of 4 s. All measurements were performed in triplicate. The relaxation data were analyzed using MultiExp Inv Analysis software, which employs a modified simultaneous iterative reconstruction technique (SIRT) algorithm for accurate relaxation time distribution under low signal-to-noise conditions[44].

## Data availability
The data that support the findings of this study are available from the corresponding author upon request. Source data are provided with this paper.

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

## Acknowledgements

K.L. acknowledges funding from the National Natural Science Foundation of China (52263012, 52563011), the project supported by "Xing Dian Ying Cai", Yunnan Fundamental Research Project (KKRD202205060, 2022), programme of Yunnan province (202101BE070001-064), and the degradable plastics engineering research center of Yunnan provincial education department (KKPU202205001). Q.Z. acknowledges support from the Swedish Foundation for International Cooperation in Research and Higher Education, STINT (CH2017-7275).

## Author contributions

Q.Z., K.L., and L.L. conceived the project. Q.Z. and K.L. supervised the project. L.L. and Y. Lu prepared samples and conducted gas adsorption analysis, SEM, mechanical test, and thermal analysis. B.L. performed X-ray scattering characterization. Z.S. performed an optical measurement. Z.S. and Y. Liu contributed to sample preparation and illustration. Q.Z. and K.L. wrote the manuscript with input from all authors.

## Funding

## Competing interests

The authors declare no competing interests.
