## [Transparent Peer Review file · Nature Communications]

Ultrastrong and Tough Paper Structure from Densified Hybrids of Multiscale Cellulose Fibers

Corresponding Author: Professor Qi Zhou

Version 0:

Reviewer comments:

Reviewer #1

(Remarks to the Author)

In this manuscript, Liqiong Liao et. al. reported an innovative approach to enhancing the mechanical properties of cellulose-based materials by hybridizing multiscale cellulose fibers. The study effectively addresses a long-standing limitation in paper-based materials, weak interfacial adhesion between fibers, by utilizing cellulose microgels and nanofibers to reinforce both microscale and nanoscale voids. The unprecedented tensile strength (811 MPa) represents the highest reported value for isotropic cellulose films to date. The innovative hybridization strategy, superior mechanical performance, energy efficiency and practical implications, adhesive performance and excellent wet strength position this material as a strong contender for replacing petrochemical-based plastics in structural and packaging applications. This is a novel and timely contribution to sustainable materials research. However, before publication in Nature Communications, the following improvements are recommended:

1. The “necking” behavior in Figure 1E suggests significant plastic deformation, which is unusual for cellulose-based materials. More discussion on whether this behavior results from fiber alignment, hydrogen bonding dynamics, or interfacial slippage would be valuable.
2. A fracture mechanics analysis (e.g., crack propagation studies, work of fracture measurements) could provide deeper insight into the material’s failure mechanisms.
3. Since bound water is a key factor in maintaining high strength, how does prolonged exposure to humid environments affect performance?
4. Previous studies focused on CNF-pulp composites. What distinguishes cellulose microgel’s role in enhancing HCP mechanics?
5. For mechanical properties, error bars or standard deviation bands should be included to show variability across samples.
6. Request for higher-resolution images of Figures 2B, 2C, and 2D to clarify the differences between the neat pulp fiber structure and the hybridized paper.
7. Why does HCP exhibit low wet strength?
8. Clarify theoretical water absorption calculation in Figure S8.
9. The study finds that a 1:1:1 ratio of pulp fibers, microgel, and B-CNF gives the best mechanical properties, but it is unclear if this is an absolute optimum. Could small adjustments in composition (e.g., 1.2:1:1 or 0.8:1:1) lead to further improvements?

Reviewer #2

(Remarks to the Author)

The manuscript by Zhou and coworkers reports on how a strong paper can be made from cellulose by mixing cellulose fibers/fibrils of different dimensions. The study builds on a large volume of work on nanopaper and various strengthening additives to paper from cellulose pulp. The paper in the current study is produced by vacuum filtration of aqueous mixtures of pulp fibres with TEMPO oxidized cellulose nanofibers and cellulose microgels. The produced papers/films displayed a high tensile strength and strain to fracture and also a good adhesion strength. The mechanical measurements are supported by SEM and WAXS to characterize the paper structure and fiber alignment. There is essentially no data supporting the discussion on the bonding mechanism and the investigated parameter space is limited.

Specific points to address:

1. Are the reported comparisons with literature data on the tensile strength and Young's modulus comparable, i.e. are the temperature and the relative humidity the same in all cases? Also, how does the density of the different materials compare? The density of the new paper presented in the manuscript needs to be determined and reported.
2. The bonding mechanisms should be investigated in detail, using e.g. spectroscopy (ssNMR, neutron spectroscopy) and MD simulations.
3. The parameter space of the films/paper needs to be expanded to better understand the critical parameters. Pulp fibers and cellulose nanofibers of different dimensions and surface chemistry should be investigated in different combinations and compositions.
4. The authors argue that the presented material could be a biodegradable alternative to plastics. What plastic should this material replace? The biodegradability of the paper needs to be determined.
5. The amount of bound water and its contribution to the interfibrillar bonding needs to be assessed and discussed.
6. Figure 4 is mixed up.

Reviewer #3

(Remarks to the Author)

The study shows a concept by using hybrid components (BC or other H-bond mediating components and fibers for cellulose-based composites) to enhance the mechanical properties which is widely used and well known. Quite often, the mechanism for promoting the mechanical properties of final products is unclear because of the use of many different components, which is exactly the case in this study. E.g. the distribution of these different components to exert their contribution to mechanical properties is already unclear. As well, more data about microgel for clear characterization is required. Moreover, to realize the adhesive properties between the hydrophilic substrates via H bonds should be common.

1. The author repeatedly mentioned in the Abstract that the material preparation is energy-efficient. Authors should quantitatively compare the energy inputs of your paper with those in other studies to support the claim of energy efficiency.
2. In Figure 1E, what do the 1, 2, 3 and three curves mean? Also, the necking indicates the decrease in local cross-sectional area, and authors can better display this decrease in the insert photograph.
3. More advanced data should be provided to support the results shown in Figure 2A. Also In Figure 3A, the explanation for the super tensile strength of cellulose paper is somewhat unclear and unsatisfactory. The explanation from the perspective of free water and hydrogen bonding of cellulose also lacks data support.
4. In Figure 2, it seems that the thickness of different samples varies greatly. Did the authors use the same method to prepare the samples (especially for the neat fibers)? Or can the authors give some discussions on the thickness results as well as related to mechanical properties?
5. In Figure 3C, the strength and other specific properties of the paper sheet are also important for this test, authors should give more information about this substrate.
6. The size of microgels and nanofibers, as well as the types and quantities of their surface functional groups, have a significant impact on capillary forces and hydrogen bonding. Such interfacial interactions represent one of the core innovative aspects of this work. I suggest the authors to provide a more detailed discussion on this topic.
7. Figure 4A: Did the author perform normalization when using NIR to compare hydrogen bond content? How is it done?
8. There are many errors in the text that need to be corrected. For example: Line 153, Figure caption of Figure 4.

Version 1:

Reviewer comments:

Reviewer #1

(Remarks to the Author)

The issues have been addressed.

Reviewer #2

(Remarks to the Author)

The authors have responded well to several points that was raised. However, the NMR study and the interpretation of the data is confusing and incomplete. The ratio of bound and free water should be determined at 70 and 90% RH, i.e. higher moisture contents. The arguments for "hydrogen bonding" are vague and needs to be supported by modeling.

Reviewer #3

(Remarks to the Author)

Authors could successfully address my concerns, which make the manuscript acceptable.

Version 2:

Reviewer comments:

Reviewer #2

(Remarks to the Author)

The addition of NMR data has addressed the remaining concerns sufficiently.

Response to the comments

We thank the reviewers for their valuable time, insightful comments and suggestions which helped us to improve our manuscript. We have made all possible efforts to address the comments and revised the manuscript accordingly. The changes made in the manuscript are highlighted in red. A point-by-point response to the comments by the reviewers is provided below.

REVIEWER COMMENTS

Reviewer #1 (Remarks to the Author):

In this manuscript, Liqiong Liao et. al. reported an innovative approach to enhancing the mechanical properties of cellulose-based materials by hybridizing multiscale cellulose fibers. The study effectively addresses a long-standing limitation in paper-based materials, weak interfacial adhesion between fibers, by utilizing cellulose microgels and nanofibers to reinforce both microscale and nanoscale voids. The unprecedented tensile strength (811 MPa) represents the highest reported value for isotropic cellulose films to date. The innovative hybridization strategy, superior mechanical performance, energy efficiency and practical implications, adhesive performance and excellent wet strength position this material as a strong contender for replacing petrochemical-based plastics in structural and packaging applications. This is a novel and timely contribution to sustainable materials research. However, before publication in Nature Communications, the following improvements are recommended:

Response: We appreciate the encouraging comments provided by the reviewer and opinion about innovativeness and quality. We have carefully considered each comment and have revised our manuscript, doing our maximum to comprehensively address all the points raised.

1. The “necking” behavior in Figure 1E suggests significant plastic deformation, which is unusual for cellulose-based materials. More discussion on whether this behavior results from fiber alignment, hydrogen bonding dynamics, or interfacial slippage would be valuable.

Response: Thanks for pointing this behavior out. Indeed, upon unloading after the onset of strain hardening but prior to fracture, the sample retained its deformed (homogeneous necking)

shape, confirming the presence of plastic deformation (Supplementary Fig. 2). We have corrected this behavior as “strain hardening” in Fig. 1e. We have updated the insert pictures in Fig. 1e and provided a movie recording tensile test of an HCP sample. We have revised the discussion as below.

Interestingly, the HCP showed a strain hardening behavior during tensile testing (insert picture in Fig. 1e and Supplementary Movie 1), similar to the plastic deformation of elastomeric polymers. Notably, upon unloading after the onset of strain hardening but prior to fracture, the sample retained its deformed (homogeneous necking) shape, confirming the presence of plastic deformation (Supplementary Fig. 2). This phenomenon was previously reported in cellulose nanopaper prepared from microfibrillated cellulose (MFC), which showed a yield stress of 91 MPa and a slope of 1.82 GPa in the plastic region. In comparison, the HCP sample demonstrated remarkably enhanced mechanical performance, with a yield stress of 305 MPa, a plastic deformation slope of 4.76 GPa, and a high strain-to-failure of 11.4%. These values suggest significantly increased frictional resistance to fiber slippage within the network and superior toughness resulting from the hybridization of multiscale cellulose fibers.

Fig. 1e Tensile stress–strain curves of the HCP prepared from the pulp fiber, microgel, and B-CNF with a composite of 1:1:1 by weight as compared to the papers prepared from the individual components; the inset photograph shows strain hardening of the HCP under tensile loading.

Supplementary Fig. 2 Photograph of the HCP sample when the stress was removed after strain hardening and before breakage, showing the plastic deformation remained.

2. A fracture mechanics analysis (e.g., crack propagation studies, work of fracture measurements) could provide deeper insight into the material's failure mechanisms.

Response: We have performed the morphological analysis of tensile fractured surface of HCP using SEM and included the discussion of fracture mechanics in the revised manuscript as below.

To elucidate the failure mechanisms underlying this behavior, the fracture surface of the HCP was examined using scanning electron microscopy (SEM; Supplementary Fig. 3). Two dominant fracture modes were identified: fiber pull-out and fiber rupture. The presence of ruptured fibers confirms that load transfer through the network reached a threshold sufficient to cause fiber breakage, enabling greater energy absorption compared to pull-out alone. This mixed-mode fracture behavior is a key contributor to the enhanced mechanical properties observed in the HCP.

Supplementary Fig. 3 SEM images of the tensile fracture surface of the HCP sample.

3. Since bound water is a key factor in maintaining high strength, how does prolonged exposure to humid environments affect performance?

Response: We have performed tensile test at higher relative humidity and added the result and discussion in the revised manuscript as below.

In addition, as RH increased to 70% and 90%, the ultimate tensile strength of the HCP decreased significantly while the strain-to-failure increased (Fig. 3d), due to higher moisture content and increased loosely bound water. These observations confirm that hydrogen bonds between cellulose and tightly bound water play a more critical role in the formation of HCP and are primarily responsible for its ultrahigh strength and toughness.

Fig. 3d Typical stress–strain curves of the HCP at different relative humidity (RH) levels.

4. Previous studies focused on CNF-pulp composites. What distinguishes cellulose microgel’s role in enhancing HCP mechanics?

Response: We have revised our manuscript to clarify the roles of B-CNF and microgel as below.

The respective roles of microgel and B-CNF in enhancing paper strength were evaluated using composite papers prepared from binary combinations of B-CNF and pulp fiber, microgel and pulp fiber, and microgel and B-CNF (Supplementary Fig. 4). When combined with pulp fibers,

B-CNF led to a modest increase in tensile strength and stiffness, attributable to its nanofibrillar structure and reinforcing effect. In contrast, the incorporation of microgel into B-CNF networks resulted in increased strain-to-failure, albeit at the expense of strength and stiffness. Notably, when microgel was combined with pulp fibers, it enhanced interfacial adhesion and contributed to a synergistic improvement in both tensile strength and toughness.

Supplementary Fig. 4 Stress-strain curves of the hybrid papers prepared from binary hybrids of (a) B-CNF and pulp fiber, (b) microgel and pulp fiber, and (c) microgel and B-CNF.

Furthermore, we have also evaluated the impact of microgel average particle size on the mechanical properties of the HCP and integrated the results in the revised manuscript as below.

The influence of microgel average particle size on the mechanical properties of the HCP was investigated. A reduction in particle size from 72.9 to 52.1 and 13.5 μm did not alter the overall shape of the stress-strain curve, but it led to a decrease in both strain-to-failure and ultimate tensile strength (Supplementary Fig. 5). This trend is analogous to the mechanical performance deterioration observed with decreasing average molar mass of cellulose.²⁹

Supplementary Fig. 5 a Size distribution of the microgels measured by Mastersizer. b Effect of the microgel size on the mechanical properties of the HCP.

5. For mechanical properties, error bars or standard deviation bands should be included to show variability across samples.

Response: We have summarized the mechanical properties data with standard deviations in Supplementary Table 2.

Supplementary Table 2 Tensile mechanical properties of the different hybrid papers presented in Fig. 1e, 1g, and 1h, Fig. 3c and 3d.

Samples ^a	Pulp fiber (SR)	Tensile Strength (MPa)	Strain at failure (%)	Young's Modulus (GPa)	Toughness (MJ/m ³)
Pulp fiber	65°	17.9±1.5	5.9±1.0	1.3±0.3	0.6±0.2
Microgel	-	102.3±7.2	3.6±0.4	5.5±0.7	2.7±1.1
B-CNF	-	327.4±15.9	2.2±0.4	20.6±2.9	4.7±0.7
HCP (1:1:1)	65°	811.0±17.1	11.4±1.0	36.1±1.8	56.5±6.1
HCP (1:1:1.2)	65°	691.2±4.0	6.6±1.1	23.7±1.0	25.5±1.6
HCP (1.2:1:1)	65°	665.2±5.0	6.2±0.7	21.3±1.0	29.1±1.3
HCP (0.8:1:1)	65°	457.2±8.0	4.8±0.4	19.9±0.6	14.6±1.0
HCP (1:1:0.8)	65°	440.6±8.9	6.2±0.5	15.6±0.9	17.6±1.0
HCP (1:1:2)	65°	38.5±3.1	9.0±1.3	3.1±0.7	2.2±0.4
HCP (1:2:1)	65°	77.8±9.1	13.2±1.3	2.9±0.2	6.6±1.1
HCP (2:1:1)	65°	55.2±3.7	12.9±3.4	1.3±0.2	4.7±0.6
HCP (1:1:1)	15°	344.2±26.1	4.0±0.5	17.0±2.0	9.3±1.2
HCP (1:1:1)	25°	437.4±27.0	4.5±1.0	22.2±2.8	13.8±1.6
HCP (1:1:1)	35°	541.4±9.0	5.5±0.5	26.0±2.6	19.0±1.8
HCP (1:1:1)	45°	585.8±14.2	5.8±0.8	29.7±0.7	22.4±1.3
HCP (1:1:1)	55°	623.7±20.5	7.5±0.5	30.5±3.3	30.7±2.0
HCP (1:1:1)	65°	811.0±17.1	11.4±1.0	36.1±1.8	56.5±6.1
HCP (1:1:1)	75°	531.8±27.2	12.0±1.0	12.5±2.2	34.4±3.8
HCP (1:1:1)					
Natural drying	65°	811.0±17.1	11.4±1.0	36.1±1.8	56.5±6.1
Completely drying	65°	157.3±15.9	5.3±1.1	8.5±0.5	4.4±0.8
RH 70%	65°	370.7±4.8	16.0±0.6	17.6±0.5	7.0±0.06
RH 90%	65°	168.0±6.7	19.9±0.8	9.4±0.7	9.2±0.3

^a The ratio in the parenthesis for each sample indicate the weight ratio of Pulp fiber, Microgel, and B-CNF.

6. Request for higher-resolution images of Figures 2B, 2C, and 2D to clarify the differences

between the neat pulp fiber structure and the hybridized paper.

Response: The higher-resolution images of Figures 2b, 2c, and 2d were added in Fig. 2 in the revised manuscript as below.

7. Why does HCP exhibit low wet strength?

Response: This is attributed to the high water absorption (550 wt%) of HCP, while the water absorption of microgel is 145 wt% (Supplementary Fig. 11). As we discussed in the manuscript, the wet tensile strength of the HCP was 1.3 MPa, while the cellulose microgel exhibited higher wet strength of 62 MPa due to lower water absorption, suggesting the interlocking structure of the cellulose microgel is robust enough to resist disassembly and water infiltration.

8. Clarify theoretical water absorption calculation in Figure S8.

Response: The theoretical water absorption of the composite film was calculated based on the weighted average of the individual components' measured water absorption values, specifically, those of pulp fiber, microgel, and B-CNF. For each component, the actual water absorption was first experimentally determined. Subsequently, the theoretical value for the composite was derived by applying the respective mass fractions of the components. We have added this clarification to the revised supplementary information.

9. The study finds that a 1:1:1 ratio of pulp fibers, microgel, and B-CNF gives the best mechanical properties, but it is unclear if this is an absolute optimum. Could small adjustments in composition (e.g., 1.2:1:1 or 0.8:1:1) lead to further improvements?

Response: Thanks for your suggestion.

We thoroughly investigated the mechanical properties of papers prepared from ternary hybrids of pulp fibers, microgel, and B-CNF at different ratios (Fig. 1g). The results revealed that small adjustments in composition (e.g. 1.2:1:1 or 0.8:1:1) could not lead to further improvements compared to the HCP at 1:1:1. The hybrid papers with higher content of pulp fibers (2:1:1) or microgel (1:2:1) showed higher strain-to-failure but significantly lower modulus and tensile strength. The HCP with a composition of 1:1:1 remarkably outperformed paper with other compositions in terms of Young's modulus, tensile strength, and toughness (Supplementary Table 2).

Fig. 1g Tensile stress–strain curves of HCPs fabricated from different weight ratios of pulp fibers, microgel, and B-CNF.

Reviewer #2 (Remarks to the Author):

The manuscript by Zhou and coworkers reports on how a strong paper can be made from cellulose by mixing cellulose fibers/fibrils of different dimensions. The study builds on a large volume of work on nanopaper and various strengthening additives to paper from cellulose pulp. The paper in the current study is produced by vacuum filtration of aqueous mixtures of pulp fibres with TEMPO oxidized cellulose nanofibers and cellulose microgels. The produced papers/films displayed a high tensile strength and strain to fracture and also a good adhesion strength. The mechanical measurements are supported by SEM and WAXS to characterize the paper structure and fiber alignment. There is essentially no data supporting the discussion on the bonding mechanism and the investigated parameter space is limited.

Response: Many thanks for your comments. To support the bonding mechanism, we have performed Low-Field Nuclear Magnetic Resonance (LF-NMR) measurements to study the spin-spin relaxation time (T_2) of water proton in the hybrid paper samples. We demonstrated the difference of adhesion mechanism between B-CNF and microgel using glass substrates. We have also expanded the parameters including adjustments in composition of HCP, the beating degree of pulp fibers, the average particle size of microgel, mechanically homogenized MFC instead of B-CNF.

Specific points to address:

1. Are the reported comparisons with literature data on the tensile strength and Young's modulus comparable, i.e. are the temperature and the relative humidity the same in all cases? Also, how does the density of the different materials compare? The density of the new paper presented in the manuscript needs to be determined and reported.

Response: Yes, they are comparable in terms of density, temperature, and relative humidity. We have added Supplementary Table 1, summarizing the mechanical properties of the hybrid cellulose paper (HCP) in our work and different studies correlated with the Ashby plot in Fig. 1f, as shown below. The densities and porosities of the papers shown in Fig. 2 were measured and summarized in Supplementary Table 3.

Supplementary Table 1 Mechanical properties of the hybrid cellulose paper (HCP) developed in this work and different studies correlated with the Ashby plot in Fig. 1f.

Sample	Tensile Strength (MPa)	Young's Modulus (GPa)	Strain at failure (%)	Density (g/cm ³)	Specific tensile strength (MPa cm ³ /g)	Specific Young's Modulus (GPa cm ³ /g)	Toughness (MJ/m ³)
Random oriented (isotropic) cellulose papers or films							
Regenerated cellulose ¹	106 ^a	4.3	12.4	1.245	85.1	3.45	9.8
BC film ^{2,3}	194 ^b	11.4	3.4	0.64	303	17.8	3.8
TEMPO-oxidized BC ⁴	102 ^a	18.5	0.9	1.19	85.7	15.5	0.6
Enz-CNF ⁵	214 ^c	13.2	10.1	1.08	198	12.2	15
Lignin-rich MFC ⁶	283 ^c	15.1	7.4	1.39	204	10.9	- ^e
Holo-CNF ⁷	320 ^c	21.0	4.8	1.47	218	14.3	9.8
TEMPO-CNF ⁸	321 ^d	17	9	1.41	228	12.1	16
MFC/wood fiber ⁹	160 ^c	10.1	4.2	0.97	165	10.4	4.4
HCP (this work)	811 ^d	36.1	11.4	1.10	737	32.8	56.5
Oriented or stretched (anisotropic) cellulose papers and films							
Denlignified wood film ¹⁰	449 ^c	51.1	1.6	1.32	340	38.7	- ^e
Regenerated cellulose ¹	253 ^a	14.6	16.5	1.245	203	11.7	41.1
BC film ²	1005 ^b	48.1	4.4	1.18	852	40.8	24.7

^a Tensile test condition: data not reported.

^b Tensile test condition: relative humidity (RH) of 40% and Temperature (T) of 23 °C.

^c Tensile test condition: RH of 50% and T of 23 °C.

^d Tensile test condition: RH of 50% and T of 25 °C.

^e Data not reported.

Supplementary Table 3 Densities of the papers of the HCP paper as compared to the papers prepared from neat pulp fiber (65° SR), neat microgel, and neat B-CNF.

Paper samples	Buck density (g/cm ³)	Porosity (%)
Pulp fibers	0.66	56
B-CNF	1.11	26
Microgel	1.15	23
Pulp fiber/B-CNF=1:1	0.84	44
Pulp fiber/Microgel=1:1	0.79	47
HCP - Pulp fiber/Microgel/B-CNF=1:1:1	1.10	27

^a Porosity is calculated according to the following equation, where ρ_{bulk} corresponds to the bulk density of the paper and $\rho_{cellulose}$ corresponds to the density for the cellulose, which is assumed to be 1.5 g/cm³.

$$Porosity(\%) = \left(1 - \frac{\rho_{bulk}}{\rho_{cellulose}}\right) \times 100$$

2. The bonding mechanisms should be investigated in detail, using e.g. spectroscopy (ssNMR, neutron spectroscopy) and MD simulations.

Response: Thanks for your suggestion. To study the effect of bound water on mechanical performance of the hybrid fibers, in addition to Near-infrared (NIR) spectroscopy, we have employed LF-NMR method to test the bonding water and free water in the HCP samples. The results, discussion and method are integrated in the revised manuscript as below.

To determine the hydration state of the bound water in the HCP sample, low field nuclear magnetic resonance (LF-NMR) measurements were performed at room temperature under relative humidity (RH) of 50% (Fig. 3b). The natural dried HCP sample had a moisture content of 5.8% and showed only a strong peak at $T_2 < 1$ ms, corresponding to tightly bound water that was mostly located in the first hydration layer, where water binds to the hydroxy groups of cellulose. After complete drying, the HCP sample had a moisture content of 7.4% and showed a significantly decreased peak for tightly bound water ($T_2 < 1$ ms) and 3 additional peaks with T_2 time of 5 ms, 40 ms, and 500 ms, respectively. These new peaks correspond to loosely bound

water that is located not far away from hydroxy groups, water confined in the pores due to capillary condensation, and bulk-like water resulting from surface condensation, respectively. These results indicate the irreversible loss of tightly bound water in HCP after completely drying.

Fig. 3b T₂ inversion spectra for the HCP prepared by natural and complete drying.

Nuclear magnetic resonance (NMR) relaxation measurements were conducted using a low-field pulsed NMR imager and analyzer (NMI20, Niumag Corporation, Shanghai, China) operating at a resonance frequency of 18 MHz. One gram of pretreated cellulose membrane powder was placed in a glass tube (15 mm inner diameter) and inserted into the NMR probe, which was maintained at a constant temperature of 32 °C. The transverse spin-spin relaxation time (T₂) was measured using the Carr-Purcell-Meiboom-Gill (CPMG) pulse sequence. The 90° and 180° pulse durations were 14 μs and 27 μs, respectively, with an inter-pulse delay (τ) of 200 μs. A total of 18,000 echoes (data points) were collected per measurement. To improve the signal-to-noise ratio, eight scans were averaged using a repetition time (TR) of 4 s. All measurements were performed in triplicate. The relaxation data were analyzed using MultiExp Inv Analysis software, which employs a modified simultaneous iterative reconstruction technique (SIRT) algorithm for accurate relaxation time distribution under low signal-to-noise conditions.⁴⁶

Regarding the adhesive bonding mechanism, we included the following demonstration in the revised manuscript.

We demonstrated the difference of adhesion mechanism between B-CNF and microgel using glass slides as the substrate as shown in Fig. 4a. The B-CNF formed scattered cellulose nanofibrils aggregates between the glass substrates due to fractures occurred in the gelled B-

CNF suspension upon drying similar to cellulose nanocrystals,⁴¹ while cellulose microgel formed a homogeneous nanofibril network without fractures owing to the high internal cohesion in the cellulose microgel upon drying.

Fig. 4a Comparison of adhesive mechanism between B-CNF and microgel for hydrophilic substrates (glass slides). 10 mg of B-CNC or microgel suspended in 1 mL of water were left to assemble between two glass plates and heated at 105 °C for 12 h. The bonded area between two glass slides were observed in bright field (scale bar: 200 μ m).

3. The parameter space of the films/paper needs to be expanded to better understand the critical parameters. Pulp fibers and cellulose nanofibers of different dimensions and surface chemistry should be investigated in different combinations and compositions.

Response: We thoroughly investigated the mechanical properties of papers prepared from ternary hybrids of pulp fibers, microgel, and B-CNF at different ratios (Fig. 1g). In addition, the beating degree of the pulp fibers has a strong impact on the mechanical properties of HCPs, which initially improved and then declined with increasing beating degree of the pulp fibers (Fig. 1h), as determined by the Schopper-Riegler ($^{\circ}$ SR) scale.

The influence of microgel average particle size on the mechanical properties of the HCP was

investigated. A reduction in particle size from 72.9 to 52.1 and 13.5 μm did not alter the overall shape of the stress-strain curve, but it led to a decrease in both strain-to-failure and ultimate tensile strength (Supplementary Fig. 5). This trend is analogous to the mechanical performance deterioration observed with decreasing average molar mass of cellulose.²⁹

Supplementary Fig. 5 a Size distribution of the microgels measured by Mastersizer. b Effect of the microgel size on the mechanical properties of the HCP.

Furthermore, mechanically homogenized MFC, derived from bleached softwood kraft pulp and characterized by widths of 10–20 nm and lengths of 5–10 μm , was used in place of B-CNF, which has a carboxylate content (1.18 mmol/g) and dimensions of 5–12.5 nm in width and 1–2.3 μm in length. Although the microgel/MFC combination showed moderate synergistic effect compared to individual components (Supplementary Fig. 6), the substitution resulted in HCP samples with lower yield stress (100 MPa) and ultimate tensile strength (210 MPa). These findings emphasize the critical role and synergistic interaction between thin nanoscale cellulose fibrils (B-CNF) and microscale cellulose microgel particles in governing the mechanical performance of the HCP.

Supplementary Fig. 6 Tensile stress–strain curves of the HCP papers prepared from the pulp fiber, microgel, and MFC with various weight ratios as compared to the papers prepared from the individual components.

4. The authors argue that the presented material could be a biodegradable alternative to plastics. What plastic should this material replace? The biodegradability of the paper needs to be determined.

Response: The high-performance HCP, with its superior mechanical properties, has the potential to serve as a sustainable alternative to conventional plastic packing straps typically made from PVC or PET.

Degradation of the HCP paper during soil burial was carried out in nature soil under ambient temperatures of 18–25 °C with soil moisture level of approximately 30–45 % in May in Kunming, Yunnan, China. The samples were buried in the soil at a depth of 5cm with gently hand-compacting. The HCP paper was biodegradable, and it showed a weight loss of 70% at 23 days as shown below.

Response Fig. 1 a The appearance of the HCP paper sample during soil burial degradation experiments. **b** The effect of soil burial on sample weight.

5. The amount of bound water and its contribution to the interfibrillar bonding needs to be assessed and discussed.

Response: Please refer to our answer to comment 2.

6. Figure 4 is mixed up.

Response: Thanks. The legend of Figure 4 was corrected, and it is now Fig. 3 in revised manuscript.

Reviewer #3 (Remarks to the Author):

The study shows a concept by using hybrid components (BC or other H-bond mediating components and fibers for cellulose-based composites) to enhance the mechanical properties which is widely used and well known. Quite often, the mechanism for promoting the mechanical properties of final products is unclear because of the use of many different components, which is exactly the case in this study. E.g. the distribution of these different components to exert their contribution to mechanical properties is already unclear. As well, more data about microgel for clear characterization is required. Moreover, to realize the adhesive properties between the hydrophilic substrates via H bonds should be common.

1. The author repeatedly mentioned in the Abstract that the material preparation is energy-efficient. Authors should quantitatively compare the energy inputs of your paper with those in other studies to support the claim of energy efficiency.

Response: We appreciate the reviewer's insightful comment. Our claim of energy efficiency is based primarily on the rapid dewatering capability of our HCP system, which significantly reduces the energy consumption typically associated with drying in conventional papermaking processes. To address this point more quantitatively, we compared the filtration time of our method for HCP to the processes reported in the literature for cellulose films prepared from TEMPO-CNF, Holo-CNF, Enz-CNF, and MFC, and summarized in Supplementary Table 4. We have added a quantitative comparison and brief discussion of this point in the revised manuscript and modified the Abstract to reflect this more precisely.

The fabrication process demonstrated notable energy efficiency, primarily due to its rapid dewatering capability, which led to significantly lower energy consumption compared to the preparation of nanocellulose-based paper.

The vacuum filtration time for the HCP was 1.417 h, significantly lower than the 7.79 h required for B-CNF, highlighting its great potential for practical applications (Supplementary Fig. 12). Moreover, the HCP exhibited superior mechanical properties to cellulose films^{29,31,32,40} prepared from TEMPO-CNF, Holo-CNF, and Enz-CNF, while also achieving a shorter filtration time and better energy savings, as shown in Fig. 3f and Supplementary Table 4.

Supplementary Table 4 Filtration time and mechanical properties of the HCP paper from this work as compared to the cellulose films prepared from TEMPO-CNF,⁸ Holo-CNF,⁷ and Enz-CNF⁵ as reported in literature.

	HCP	TEMPO-CNF ⁸	Holo-CNF ⁷	Enz-CNF ⁵
Filtration time (h)	< 1.5	~ 24	< 2.5	< 2.5
Thickness (μm)	70	30–40	35	60–80
Tensile strength (MPa)	811	321	320	214
Tensile modulus (GPa)	36.1	17	21	13.2
Strain at break (%)	11.4	9	4.8	10.1
Toughness (MJ/m^3)	56.5	16	9.8	15
Specific strength ($\text{MPa cm}^3/\text{g}$)	737	228	218	198
Specific modulus ($\text{GPa cm}^3/\text{g}$)	32.8	12.1	14.3	12.2

2. In Figure 1E, what do the 1, 2, 3 and three curves mean? Also, the necking indicates the decrease in local cross-sectional area, and authors can better display this decrease in the insert photograph.

Response: Thank you for raising this question. The three curves labeled 1, 2, and 3 in Fig. 1e represent the tensile stress–strain profiles of three independent specimens of the HCP with a component ratio of 1:1:1. These curves illustrate the reproducibility and variability of the mechanical behavior under standardized tensile testing conditions. Upon unloading after the onset of strain hardening but prior to fracture, the sample retained its deformed (homogeneous necking) shape, confirming the presence of plastic deformation (Supplementary Fig. 2). We have corrected this behavior as “strain hardening” in Fig. 1e. We have updated the insert pictures in Fig. 1e and provided a movie recording tensile test of an HCP sample. We have revised the discussion as below.

Interestingly, the HCP showed a strain hardening behavior during tensile testing (insert picture in Fig. 1e and Supplementary Movie 1), similar to the plastic deformation of elastomeric

polymers. Notably, upon unloading after the onset of strain hardening but prior to fracture, the sample retained its deformed (homogeneous necking) shape, confirming the presence of plastic deformation (Supplementary Fig. 2). This phenomenon was previously reported in cellulose nanopaper prepared from microfibrillated cellulose (MFC), which showed a yield stress of 91 MPa and a slope of 1.82 GPa in the plastic region. In comparison, the HCP sample demonstrated remarkably enhanced mechanical performance, with a yield stress of 305 MPa, a plastic deformation slope of 4.76 GPa, and a high strain-to-failure of 11.4%. These values suggest significantly increased frictional resistance to fiber slippage within the network and superior toughness resulting from the hybridization of multiscale cellulose fibers.

Fig. 1e Tensile stress–strain curves of the HCP prepared from the pulp fiber, microgel, and B-CNF with a composite of 1:1:1 by weight as compared to the papers prepared from the individual components; the inset photograph shows strain hardening of the HCP under tensile loading.

Supplementary Fig. 2 Photograph of the HCP sample when the stress was removed after strain hardening and before breakage, showing the plastic deformation remained.

3. More advanced data should be provided to support the results shown in Figure 2A. Also In Figure 3A, the explanation for the super tensile strength of cellulose paper is somewhat unclear and unsatisfactory. The explanation from the perspective of free water and hydrogen bonding of cellulose also lacks data support.

Response: To study the effect of bound water on mechanical performance of the hybrid fibers, in addition to Near-infrared (NIR) spectroscopy, we have employed LF-NMR method to test the bonding water and free water in the HCP samples.

To determine which type of hydrogen bonding primarily influences mechanical properties, the HCP was completely dried under vacuum at 110 °C for approximately 2 to 3 hours to remove bound water. The NIR spectra (**Fig. 3a**) revealed a significant difference between HCP subjected to complete drying and natural drying, marked by the absence of the peak at 5178 cm^{-1} , which corresponds to bound water.³⁹ To determine the hydration state of the bound water in the HCP sample, low field nuclear magnetic resonance (LF-NMR) measurements were performed at room temperature under relative humidity (RH) of 50% (Fig. 3b). The natural dried HCP sample had a moisture content of 5.8% and showed only a strong peak at $T_2 < 1$ ms, corresponding to tightly bound water that was mostly located in the first hydration layer, where water binds to the hydroxy groups of cellulose. After complete drying, the HCP sample had a moisture content of 7.4% and showed a significantly decreased peak for tightly bound water ($T_2 < 1$ ms) and 3 additional peaks with T_2 time of 5 ms, 40 ms, and 500 ms, respectively. These new peaks correspond to loosely bound water that is located not far away from hydroxy groups, water confined in the pores due to capillary condensation, and bulk-like water resulting from surface condensation, respectively. These results indicate the irreversible loss of tightly bound water in HCP after completely drying. The mechanical properties of HCP declined drastically after complete drying, with tensile strength decreasing from 811 MPa to 167 MPa (Fig. 3c). In addition, as RH increased to 70% and 90%, the ultimate tensile strength of the HCP decreased significantly while the strain-to-failure increased (Fig. 3d), due to higher moisture content and increased loosely bound water. These observations confirm that hydrogen bonds between cellulose and tightly bound water play a more critical role in the formation of HCP and are primarily responsible for its ultrahigh strength and toughness.

Fig. 3: Influence of bound water on the performance of hybrid fibers. a Near-infrared (NIR) spectra of the HCP paper after complete and natural drying. **b** T_2 inversion spectra for the HCP prepared by natural and complete drying. **c** Representative stress–strain curves of the HCP under both drying conditions. **d** Typical stress–strain curves of the HCP at different relative humidity (RH) levels.

4. In Figure 2, it seems that the thickness of different samples varies greatly. Did the authors use the same method to prepare the samples (especially for the neat fibers)? Or can the authors give some discussions on the thickness results as well as related to mechanical properties?

Response: Indeed, the thickness of different samples varies as they have different densities when we used the same method to prepare different samples including neat fibers, binary fibers, and ternary fibers. Their densities and porosities are summarized in supplementary Table 3, and we discussed the porosity and mechanical properties as below.

As shown in Supplementary Table 3, the HCP paper exhibited porosity comparable to that of neat B-CNF and aerogel papers yet demonstrated markedly superior mechanical performance. This enhancement suggests strong interfacial adhesion and improved interfacial toughness

arising from the synergistic hybridization of pulp fibers with B-CNF and microgel.

Supplementary Table 3 Densities of the papers of the HCP paper as compared to the papers prepared from neat pulp fiber (65° SR), neat microgel, and neat B-CNF.

Paper samples	Buck density (g/cm ³)	Porosity (%)
Pulp fibers	0.66	56
B-CNF	1.11	26
Microgel	1.15	23
Pulp fiber/B-CNF=1:1	0.84	44
Pulp fiber/Microgel=1:1	0.79	47
HCP - Pulp fiber/Microgel/B-CNF=1:1:1	1.10	27

^a Porosity is calculated according to the following equation, where ρ_{bulk} corresponds to the bulk density of the paper and $\rho_{cellulose}$ corresponds to the density for the cellulose, which is assumed to be 1.5 g/cm³.

$$Porosity(\%) = \left(1 - \frac{\rho_{bulk}}{\rho_{cellulose}}\right) \times 100$$

We also prepared HCP samples with different thickness, 40, 70, and 100 μm . The mechanical properties of samples with thickness of 40 and 70 μm were similar, while the sample with even higher thickness (100 μm) showed slightly higher plastic deformation slope and lower strain-to-failure, as shown below. In general, the HCP samples are homogeneous and defects free in these different thicknesses.

Response Fig. 2 Stress-strain curves of HCP with different thickness.

5. In Figure 3C, the strength and other specific properties of the paper sheet are also important for this test, authors should give more information about this substrate.

Response: Thanks for raising this point. The paper sheets for the adhesive test were the control paper prepared from the neat pulp fibers (65°SR).

6. The size of microgels and nanofibers, as well as the types and quantities of their surface functional groups, have a significant impact on capillary forces and hydrogen bonding. Such interfacial interactions represent one of the core innovative aspects of this work. I suggest the authors to provide a more detailed discussion on this topic.

Response: Thanks for your suggestion. We have provided further detailed discussion on the topic in the revised manuscript as below.

The influence of microgel average particle size on the mechanical properties of the ternary hybrids was further investigated. A reduction in particle size from 72.9 to 52.1 and 13.5 μm did not alter the overall shape of the stress-strain curve, but it led to a decrease in both strain-to-failure and ultimate tensile strength (Supplementary Fig. 5). This trend is analogous to the mechanical performance deterioration observed with decreasing average molar mass of cellulose.²⁹ Furthermore, mechanically homogenized MFC, derived from bleached softwood kraft pulp and characterized by widths of 10–20 nm and lengths of 5–10 μm , was used in place of B-CNF, which has a carboxylate content (1.18 mmol/g) and dimensions of 5–12.5 nm in width and 1–2.3 μm in length. Although the microgel/MFC combination showed moderate synergistic effect compared to individual components (Supplementary Fig. 6), the substitution resulted in HCP samples with lower yield stress (100 MPa) and ultimate tensile strength (210 MPa). These findings emphasize the critical role and synergistic interaction between thin nanoscale cellulose fibrils (B-CNF) and microscale cellulose microgel particles in governing the mechanical performance of the HCP.

Supplementary Fig. 5 a Size distribution of the microgels measured by Mastersizer. b Effect of the microgel size on the mechanical properties of the HCP.

Supplementary Fig. 6 Tensile stress–strain curves of the HCP papers prepared from the pulp fiber, microgel, and MFC with various weight ratios as compared to the papers prepared from the individual components.

7. Figure 4A: Did the author perform normalization when using NIR to compare hydrogen bond content? How is it done?

Response: All near-infrared (NIR) spectra were normalized using the min–max normalization method, implemented in the OPUS software. The normalization was carried out according to the following formula:

$$\text{Normalized spectrum} = \frac{\textit{Original spectrum} - \textit{Minimum value}}{\textit{Maximum value} - \textit{Minimum value}}$$

This approach ensures that the spectral data are scaled uniformly, facilitating accurate comparison and analysis.

8. There are many errors in the text that need to be corrected. For example: Line 153, Figure caption of Figure 4.

Response: We carefully revised the manuscript and corrected the errors in the text including the caption of Figure 4 (currently Fig. 3 in the revised manuscript).

Response to the comments

REVIEWER COMMENTS

Reviewer #1 (Remarks to the Author):

The issues have been addressed.

Response: Thank you.

Reviewer #2 (Remarks to the Author):

The authors have responded well to several points that was raised. However, the NMR study and the interpretation of the data is confusing and incomplete. The ratio of bound and free water should be determined at 70 and 90% RH, i.e. higher moisture contents. The arguments for "hydrogen bonding" are vague and needs to be supported by modeling.

Response: Thank you very much for your valuable comments on the NMR analysis in “influence of bound water on mechanical performance of the hybrid fibers”. In response, we performed additional low-field NMR (LF-NMR) measurements on naturally dried HCP samples conditioned at 70% and 90% relative humidity (RH) and further reconditioned the samples back to 50% RH to evaluate the recovery of mechanical properties.

These new results confirm that tightly bound water, which is hydrogen-bonded to cellulose chains, plays a key structural role in the fiber network. This interpretation is supported by previously reported simulation studies, and we have now added the relevant citations. The new LF-NMR results are presented in Fig. 3b and the corresponding mechanical property recovery results in Fig. 3d. The Discussion section has been revised accordingly in the updated manuscript.

For clarity, the revised text has also been included below.

Fig. 3: Influence of bound water on mechanical performance of the hybrid fibers. b T_2 inversion spectra for the HCP prepared by natural and complete drying and conditioned at different relative humidity (RH) levels. **d** Typical stress–strain curves of the natural dried HCP at different RH levels.

To determine the hydration state of the bound water in the HCP sample, low field nuclear magnetic resonance (LF-NMR) measurements were performed at room temperature (Fig. 3b). The natural dried HCP sample had a moisture content of 5.8% under 50% relative humidity (RH) and showed only a strong peak at $T_2 < 1$ ms, corresponding to tightly bound water that was mostly located in the first hydration layer, where water binds to the hydroxy groups of cellulose. When the HCP samples were conditioned under 70% and 90% RH, their moisture contents increased to 10.8% and 18.5%, respectively. Correspondingly, the T_2 relaxation times of the bound water shifted to values above 1 ms and exhibited broader distributions, indicating a greater contribution from more loosely bound water under high humidity conditions. Additionally, small relaxation peaks appeared at 70 ms and 500 ms, which suggests the presence of a minor fraction of free or bulk-like water at high RH levels. After complete drying, the HCP sample had a moisture content of 7.4% under 50% RH and showed a significantly decreased peak for tightly bound water ($T_2 < 1$ ms) and 3 additional peaks with T_2 time of 5 ms, 40 ms, and 500 ms, respectively. These new peaks correspond to loosely bound water that is located not far away from hydroxy groups, as well as free water confined in the pores due to capillary condensation and bulk-like water resulting from surface condensation, respectively. These results indicate the irreversible loss of tightly bound water in HCP after completely drying. The mechanical properties of HCP declined drastically after complete drying, with tensile strength decreasing from 811 MPa to 167 MPa (Fig. 3c). Furthermore, when the RH was increased to 70% and 90%, the ultimate tensile strength of the natural dried HCP decreased significantly

while the strain-to-failure increased (Fig. 3d), consistent with the higher moisture content and the greater proportion of loosely bound water (Fig. 3b). Notably, the tensile strength of the natural dried HCP was largely restored after reconditioning the samples back to 50% RH, indicating that the humidity-induced changes were reversible. These observations indicate that tightly bound water, which is hydrogen-bonded to the cellulose chains,^{40,41} plays a key structural role in HCP. This tightly bound water contributes to the dense packing and load transfer within the network of multiscale cellulose fibers and is therefore a primary factor underlying the material's ultrahigh strength and toughness.

Reviewer #3 (Remarks to the Author):

Authors could successfully address my concerns, which make the manuscript acceptable.

Response: Thank you